# Combination of Muscle Quantity and Quality Is Useful to Assess the Necessity of Surveillance after a 5-Year Cancer-Free Period in Patients Who Undergo Radical Cystectomy: A Multi-Institutional Retrospective Study

**DOI:** 10.3390/cancers15051489

**Published:** 2023-02-27

**Authors:** Naoki Fujita, Masaki Momota, Hirotaka Horiguchi, Itsuto Hamano, Jotaro Mikami, Shingo Hatakeyama, Hiroyuki Ito, Takahiro Yoneyama, Yasuhiro Hashimoto, Shoji Nishimura, Kazuaki Yoshikawa, Chikara Ohyama

**Affiliations:** 1Department of Urology, Hirosaki University Graduate School of Medicine, 5 Zaifucho, Hirosaki 036-8562, Japan; 2Department of Urology, Mutsu General Hospital, 1-2-8 Kogawamachi, Mutsu 035-8601, Japan; 3Department of Urology, Hakodate Municipal Hospital, 1-10-1 Minatomachi, Hakodate 041-8680, Japan; 4Department of Urology, Towada City Central Hospital, 14-8 Nishijyunibancho, Towada 034-0093, Japan; 5Department of Advanced Blood Purification Therapy, Hirosaki University Graduate School of Medicine, 5 Zaifucho, Hirosaki 036-8562, Japan; 6Department of Urology, Aomori Rosai Hospital, 1 Minamigaoka Shiroganemachi, Hachinohe 031-8551, Japan; 7Department of Advanced Transplant and Regenerative Medicine, Hirosaki University Graduate School of Medicine, 5 Zaifucho, Hirosaki 036-8562, Japan; 8Department of Urology, Aomori City Hospital, 1-14-20 Katsuta, Aomori 030-0821, Japan

**Keywords:** bladder cancer, radical cystectomy, sarcopenia, psoas muscle index, myosteatosis, intramuscular adipose tissue content, surveillance

## Abstract

**Simple Summary:**

Although continuous surveillance after a 5-year cancer-free period in patients with bladder cancer who undergo curative surgery is recommended, optimal candidates for continuous surveillance remain unclear. Sarcopenia is associated with an unfavorable prognosis in bladder cancer. We aimed to investigate the impact of low muscle quantity and quality (defined as severe sarcopenia) on prognosis after a 5-year cancer-free period in patients who underwent radical cystectomy. Our results showed that the 10-year recurrence rate after a 5-year cancer-free period was low (approximately 5%), and severe sarcopenia was not associated with increased recurrence risk. Moreover, severe sarcopenia was selected as a significant risk factor for mortality unrelated to bladder cancer. Taken together, patients with severe sarcopenia might not need continuous surveillance after a 5-year cancer-free period, considering high mortality unrelated to bladder cancer.

**Abstract:**

Background: Although continuous surveillance after a 5-year cancer-free period in patients with bladder cancer (BC) who undergo radical cystectomy (RC) is recommended, optimal candidates for continuous surveillance remain unclear. Sarcopenia is associated with unfavorable prognosis in various malignancies. We aimed to investigate the impact of low muscle quantity and quality (defined as severe sarcopenia) on prognosis after a 5-year cancer-free period in patients who underwent RC. Methods: We conducted a multi-institutional retrospective study assessing 166 patients who underwent RC and had five years or more of follow-up periods after a 5-year cancer-free period. Muscle quantity and quality were evaluated using the psoas muscle index (PMI) and intramuscular adipose tissue content (IMAC) using computed tomography images five years after RC. Patients with lower PMI and higher IMAC values than the cut-off values were diagnosed with severe sarcopenia. Univariable analyses were performed to assess the impact of severe sarcopenia on recurrence, adjusting for the competing risk of death using the Fine-Gray competing risk regression model. Moreover, the impact of severe sarcopenia on non-cancer-specific survival was evaluated using univariable and multivariable analyses. Results: The median age and follow-up period after the 5-year cancer-free period were 73 years and 94 months, respectively. Of 166 patients, 32 were diagnosed with severe sarcopenia. The 10-year RFS rate was 94.4%. In the Fine-Gray competing risk regression model, severe sarcopenia did not show a significant higher probability of recurrence, with an adjusted subdistribution hazard ratio of 0.525 (*p* = 0.540), whereas severe sarcopenia was significantly associated with non-cancer-specific survival (hazard ratio 1.909, *p* = 0.047). These results indicate that patients with severe sarcopenia might not need continuous surveillance after a 5-year cancer-free period, considering the high non-cancer-specific mortality.

## 1. Introduction

Bladder cancer (BC) is the tenth most common cancer worldwide [1]. Radical cystectomy (RC) with pelvic lymph node dissection and urinary diversion remains the gold standard treatment for muscle-invasive and high-risk non-muscle-invasive BC [2].

Continuous surveillance after a 5-year cancer-free period in patients who undergo RC is recommended by the European Association of Urology guidelines [2]. However, late recurrence after RC has been reported to be infrequent [3,4]. Moreover, optimal candidates for continuous surveillance remain unclear. Identifying these candidates may be helpful in the development of individualized surveillance protocols.

Sarcopenia is represented by two dysregulation patterns of body composition: loss of skeletal muscle quantity (myopenia) and quality (myosteatosis) [5]. Although sarcopenia has been reported to be associated with unfavorable prognosis in patients who underwent surgical treatment for various malignancies [6,7], the prognostic value of sarcopenia in patients who undergo RC remains controversial [8,9,10,11]. Moreover, no study has investigated its impact on oncological outcomes and non-cancer-specific mortality after a 5-year cancer-free period. Considering the close relationship between sarcopenia and high mortality caused by non-malignant diseases [12,13,14], we hypothesized that patients with low muscle quantity and quality (defined here as severe sarcopenia) might have a high non-cancer-specific mortality; therefore, they might not need continuous surveillance after a 5-year cancer-free period.

The aim of the present study was to evaluate the impact of low muscle quantity and quality on recurrence-free survival (RFS) and non-cancer-specific survival after a 5-year cancer-free period in patients with BC who underwent RC.

## 2. Materials and Methods

### 2.1. Ethics Statement

This study followed the principles of the Declaration of Helsinki and was approved by the ethics committees of the Hirosaki University Graduate School of Medicine (authorization number: 2019-099-1) and all hospitals included in this study. Written consent was not obtained due to the public disclosure of the study information (opt-out approach).

### 2.2. Patient Selection

To include patients who had sufficient follow-up periods (five years or more) after a 5-year cancer-free period, we retrospectively evaluated 431 patients with BC who underwent RC between October 1995 and December 2012 at one academic center and five general hospitals. We excluded 193 patients who experienced local recurrence and/or distant metastasis, died from any cause, or were lost to follow-up within five years after RC and 72 patients who had no information on their heights or digital computed tomography (CT) scans available for body composition analysis. Ultimately, 166 patients were included in this study (Figure 1).

### 2.3. Evaluation of Variables

The following variables were analyzed: age, sex, Eastern Cooperative Oncology Group performance status (ECOG PS), hypertension (HTN), diabetes mellitus, history of cardiovascular disease (CVD), chronic kidney disease (CKD), clinical stage, neoadjuvant chemotherapy (NAC), urinary diversion, pathological outcomes, and adjuvant chemotherapy. Age and comorbidities at five years after RC were used in the analyses. Renal function was evaluated by estimated glomerular filtration rate (eGFR) using a modified version of the abbreviated Modification of Diet in Renal Disease Study formula for Japanese patients [15] and CKD was defined as eGFR < 60 mL/min/1.73 m^2^. Tumor stage was assigned according to the 2009 TNM classification system recorded by the Union of International Cancer Control. Tumor grade was classified according to the 1973 World Health Organization classification system.

### 2.4. NAC and Adjuvant Chemotherapy

Since September 2004, patients have received two–four courses of NAC, composed of a platinum-based combination regimen of gemcitabine plus cisplatin, gemcitabine plus carboplatin, or methotrexate, vinblastine, adriamycin, and cisplatin. Regimens were selected based on guidelines regarding eligibility for the proper use of cisplatin, overall patient status, and the clinician’s discretion. The cycles were repeated every 21 days.

Adjuvant chemotherapy was not routinely administered. Indications for adjuvant chemotherapy included pT4, pathological lymph node involvement, grade 3, lymphovascular invasion, or positive surgical margins in patients who were not treated with NAC. Patients were selected for adjuvant chemotherapy at the clinician’s discretion. We administered one–three courses of adjuvant chemotherapy to patients with a feasible postoperative status for toxic chemotherapy. Adjuvant chemotherapy comprises a platinum-based combination regimen of gemcitabine plus cisplatin, gemcitabine plus carboplatin, or methotrexate, vinblastine, doxorubicin, and cisplatin.

### 2.5. Surgical Procedures

RC was performed using a previously described basic technique [16]. Briefly, the patients underwent RC, standard pelvic lymph node dissection, and urinary diversion (orthotopic ileal neobladder construction, ileal conduit diversion, or cutaneous ureterostomy).

### 2.6. Follow-Up Schedule 

The follow-up schedule after the 5-year cancer-free period comprised annual urine analysis, urine cytology, blood chemistry, and lung, abdominal, and pelvic CT scans.

### 2.7. Evaluation of Muscle Quantity and Quality

Muscle quantity was evaluated using the psoas muscle index (PMI). We measured the cross-sectional areas of the right and left psoas muscles on plain CT images at the level of the third lumbar vertebra (L3) five years after RC. The muscles were identified based on their anatomical features, and the bilateral psoas muscle areas were evaluated using manual tracing. The PMI was calculated by normalizing these cross-sectional areas to their height (cm^2^/m^2^) [17].

Muscle quality was evaluated based on intramuscular adipose tissue content (IMAC) using L3 level plain CT images five years after RC. We precisely traced the multifidus muscle and subcutaneous fat to measure their CT values (Hounsfield units). IMAC was calculated by dividing the CT value of the multifidus muscles by that of the subcutaneous fat. A higher IMAC indicates a greater amount of adipose tissue in the skeletal muscles and, therefore, lower skeletal muscle quality [18].

Since the ranges of PMI and IMAC in men and women are quite different [19,20], and their optimal cut-off values for mortality in patients with BC have been to be established, their optimal cut-off values for non-cancer-specific mortality were calculated separately for men and women using receiver operating characteristic (ROC) curves. In the present study, we defined patients with both low muscle quantity and quality as patients with severe sarcopenia. Patients were divided into two groups: those with lower PMI and higher IMAC values than their cut-off values (severe sarcopenia group) and those with higher PMI and/or lower IMAC values (control group) (Figure 1).

### 2.8. Statistical Analysis

SPSS version 24.0 (SPSS Corp., Armonk, NY, USA), R 4.0.2 (The R Foundation for Statistical Computing, Vienna, Austria), and GraphPad Prism 5.03 (GraphPad Software, San Diego, CA, USA) were used for statistical analyses. Quantitative variables are expressed as medians with interquartile ranges. Differences in quantitative variables between the two groups were analyzed using the Mann–Whitney U test. Categorical variables were compared using Fisher’s exact test or the chi-squared test. RFS, overall survival (OS), and non-cancer-specific survival were evaluated using the Kaplan–Meier method and compared using the log-rank test. Moreover, the cumulative incidences of recurrence was estimated and death before recurrence was defined as a competing risk. The Gray test was performed to compare cumulative incidences between the control and severe sarcopenia groups. Subsequent univariable analyses were performed to assess the impact of severe sarcopenia on recurrence, adjusting for the competing risk of death using the Fine-Gray subdistribution hazards model. Univariable Cox proportional hazards regression analyses were performed to identify the significant factors associated with RFS. Univariable and multivariable Cox proportional hazards regression analyses were performed to evaluate the impact of severe sarcopenia on non-cancer-specific survival. These outcomes were calculated from five years after RC to the date of the first event or last follow-up. Recurrence was defined as local pelvic recurrence, remnant urothelial recurrence, or distant metastasis. Non-cancer-specific mortality was defined as death unrelated to BC. Statistical significance was set at *p* < 0.05. 

## 3. Results

### 3.1. Patients’ Backgrounds

The median age and follow-up period after the 5-year cancer-free period were 73 years and 94 months, respectively. Of the 166 patients, 85 (51%) and 19 (11%) received NAC and adjuvant chemotherapy, respectively. The patients’ backgrounds are summarized in Table 1.

### 3.2. Evaluation of Muscle Quantity and Quality

The median PMI values in men and women were 6.18 cm^2^/m^2^ and 4.65 cm^2^/m^2^, respectively. The optimal cut-off values of PMI for non-cancer-specific mortality in men and women was 5.28 cm^2^/m^2^ and 6.35 cm^2^/m^2^, respectively. Of the 166 patients, 94 (57%) and 72 (43%) had PMI values higher and lower than the cut-off values, respectively.

The median IMAC values in men and women were −0.46 and −0.33, respectively. The optimal cut-off values of IMAC for non-cancer-specific mortality in men and women was −0.49 and −0.04, respectively. Of the 166 patients, 84 (51%) and 82 (49%) had IMAC values lower and higher than the cut-off values, respectively.

Patients were divided into two groups: those with lower PMI and higher IMAC values (severe sarcopenia group, *n* = 32) and those with higher PMI and/or lower IMAC values (control group, *n* = 134) (Figure 1). No significant differences in patients’ background were observed between the two groups, except for age and ECOG PS (Table 1).

### 3.3. BC Recurrence

By the end of the follow-up period after the 5-year cancer-free period, nine (5.4%) patients experienced BC recurrence, including recurrence in the upper urinary tract (*n* = 3), lymph nodes (*n* = 2), urethra (*n* = 1), local pelvis (*n* = 1), neobladder (*n* = 1), and distant metastasis (*n* = 1). Of the nine patients who experienced BC recurrence, eight (6.0%) and one (3.1%) patients were in the control and severe sarcopenia groups, respectively. The 5-year and 10-year RFS rates were 95.2% and 94.4%, respectively (Figure 2A). Almost all recurrence detection rates ([number of patients with recurrence/number of patients with surveillance during a certain period] × 100) were under 1% throughout the entire follow-up period (Figure 2B).

In the Gray test, the cumulative incidence rate of recurrence was not significantly different between the control and severe sarcopenia groups (Figure 2C, *p* = 0.528). In the univariable analyses, none of the patient factors, clinical stage, perioperative chemotherapy, or pathological outcomes were significantly associated with shorter RFS (Appendix A). Similarly, none of lower PMI, higher IMAC, and severe sarcopenia showed significant higher probabilities of recurrence (Figure 2D; subdistribution hazard ratio [SHR] 0.664, 95% confidence interval [CI] 0.168–2.630, *p* = 0.560; SHR 0.814, 95% CI 0.220–3.010, *p* = 0.760; SHR 0.525, 95% CI 0.067–4.140, *p* = 0.540; respectively).

### 3.4. OS and Non-Cancer-Specific Survival

By the 5-year, 10-year, and end of the follow-up period after the 5-year cancer-free period, 29 (18%), 51 (31%), and 64 (39%) patients died from any cause, respectively. The main causes of death during the entire follow-up period were other malignancies (25%) and CVD (22%), followed by infectious diseases (14%) (Figure 3). Of the 64 patients who died from any cause, six (9.4%) died of BC (Figure 3).

The OS in patients with lower PMI values was significantly shorter than that in patients with higher PMI values (Figure 4A, *p* = 0.003). The OS in patients with higher IMAC values was significantly shorter than that in patients with lower IMAC values (Figure 4B, *p* = 0.025). The OS in the severe sarcopenia group was significantly shorter than that in the control group (Figure 4C, *p* < 0.001).

By the end of the follow-up period after the 5-year cancer-free period, 37 (28%) and 21 (66%) patients in the control and severe sarcopenia group died from non-cancer-specific cause, respectively. The 5-year and 10-year non-cancer-specific mortality rates in the severe sarcopenia group were significantly higher than those in the control group (Figure 5A,B; 38% vs. 10%, *p* < 0.001; 56% vs. 21%, *p* < 0.001; respectively). The non-cancer-specific survival in patients with lower PMI values was significantly shorter than that in patients with higher PMI values (Figure 5C, *p* = 0.001). The non-cancer-specific survival in patients with higher IMAC values was significantly shorter than that in patients with lower IMAC values (Figure 5D, *p* = 0.010). The non-cancer-specific survival in the severe sarcopenia group was significantly shorter than that in the control group (Figure 5E, *p* < 0.001). In univariable analyses, age, EGOS PS, HTN, CKD, lower PMI, higher IMAC, and severe sarcopenia were significantly associated with shorter non-cancer-specific survival (Table 2). In multivariable analyses, lower PMI and higher IMAC were not significantly associated with shorter non-cancer-specific survival (HR 1.267, 95% CI 0.696–2.308, *p* = 0.439; HR 1.377, 95% CI 0.688–2.757, *p* = 0.367; respectively), whereas severe sarcopenia was significantly associated with shorter non-cancer-specific survival (HR 1.909, 95% CI 1.007–3.619, *p* = 0.047) (Table 3). Age and CKD were also associated with shorter non-cancer-specific survival (Table 3).

## 4. Discussion

To the best of our knowledge, this is the first study to evaluate the impact of low muscle quantity and quality (defined here as severe sarcopenia) on oncological outcomes and non-cancer-specific mortality after a 5-year cancer-free period in patients with BC who underwent RC. The results of the present study showed that the 10-year recurrence rate after the 5-year cancer-free period was low (approximately 5%), and severe sarcopenia was not associated with increased recurrence risk. In contrast, severe sarcopenia was identified as a significant risk factor for non-cancer-specific mortality. These results suggest that patients with severe sarcopenia may not need continuous surveillance after a 5-year cancer-free period. Although a prospective validation study with a larger sample size is warranted, these results might be helpful for clinicians to optimize individualized surveillance protocols after a 5-year cancer-free period.

In the present study, neither severe sarcopenia nor low muscle quantity or quality was associated with BC recurrence after a 5-year cancer-free period. Although several studies have investigated the impact of preoperative sarcopenia on oncological outcomes in patients who underwent RC [8,9,10,11], to our knowledge, there is no available evidence about its impact on oncological outcomes after a 5-year cancer-free period in both BC and other malignancies. However, our results are consistent with those of previous studies that have focused on preoperative sarcopenia. Smith et al. revealed that sarcopenia evaluated by total psoas area was not associated with worse 2-year survival in 200 patients with BC who underwent RC [10]. Likewise, Wang et al. demonstrated no significant association between sarcopenia and shorter disease-free survival in 112 patients with BC who underwent RC [21]. In contrast, Ornaghi et al. reported opposite results. They conducted a systematic literature review to investigate the impact of sarcopenia on long-term mortality rates in patients with BC treated with RC and revealed that sarcopenia was significantly associated with unfavorable 5-year cancer-specific survival (CSS) (HR 1.73, *p* < 0.05) [22]. Similarly, a systematic review and meta-analysis conducted by Hu et al. demonstrated that sarcopenia was associated with poor CSS in patients with BC who underwent RC (HR 1.73, *p* < 0.001) [23]. Although we do not have a clear answer about our negative results, these conflicting results might be caused by the varied definitions of sarcopenia between studies due to the lack of international consensus. Because the lack of available evidence and several limitations in the present study, especially the small number of recurrence events, prevent us from making definitive conclusions, further prospective studies with an appropriate sample size and recurrence events are warranted.

In the present study, severe sarcopenia (low muscle quantity and quality) was associated with increased non-cancer-specific mortality after a 5-year cancer-free period, whereas low muscle quantity or quality alone had marginal effects. Although many studies have investigated the impact of both low muscle quantity and quality on prognosis in a single study of several malignant and non-malignant diseases [5,24,25,26], the combined effects of these parameters have rarely been reported. Hopkins et al. assessed 968 patients with colorectal cancer who underwent curative resection and demonstrated that both low muscle quantity and low muscle radiodensity were independently predictive of worse OS (HR 1.45 and HR 1.53, respectively), but the presence of both increased the HR for OS (HR 2.23) [27]. Similarly, Caan et al. assessed 1628 female patients with colorectal cancer who underwent surgical resection and revealed that patients with both low muscle quantity and high total adipose tissue area had a higher risk of overall mortality (HR 1.64) than patients with low muscle quantity or high total adipose tissue area alone (HR 1.38 and HR 1.30, respectively) [28]. Although the included patient populations and evaluated outcomes in these studies were different from those in the present study, these results indicate the potential additive effects of low muscle quantity and quality on prognosis in patients with malignancies.

It is unclear how other diseases contribute to the mortality of BC survivors after RC. In the present study, only six (3.6%) patients died from BC after the 5-year cancer-free period, whereas 58 (34.9%) died from other causes (Figure 3). Kong et al. reported similar results. They assessed 81,843 patients with BC who survived 5–10 years after treatment (93.9% were treated with surgery) and demonstrated that only 6.9% of them died from BC while 47.9% died from other causes, including CVD (11.0%), pulmonary disease (7.7%), and other cancers (3.0%) [29]. Moreover, late recurrence after RC has been reported to be infrequent [3,4]. These results indicate that the contribution of other diseases to mortality after a 5-year cancer-free period is much greater than that of BC. The association between sarcopenia and increased mortality caused by CVD and infectious diseases has been reported [12,13,14,30]. Moreover, our results showed a relationship between sarcopenia and increased non-cancer-specific mortality after a 5-year cancer-free period. Taken together, patients with sarcopenia might not need continuous surveillance after a 5-year cancer-free period in patients who undergo RC.

The present study had several limitations. First, we were unable to control for selection bias and other unquantifiable confounders in retrospective studies. Moreover, patients without available CT scans for muscle quantity and quality measurements were excluded, which might have caused a selection bias. In addition, skeletal muscle loss is associated with aging and patients in the severe sarcopenia group were significantly older than patients in the control group in the present study (Table 1). Thus, it might have caused an association bias regardless of the adjustment for age in the multivariable analyses. Second, a relatively small number of patients were enrolled, and the number of recurrence events was also small. Moreover, the small number of cancer-specific deaths prevented us from evaluating cancer-specific survival. Third, sarcopenia was assessed by manual tracing, which may have been subject to human error. Finally, given its retrospective nature, we had no information on other frailty metrics, such as walking speed, grip strength, and nutritional status.

## 5. Conclusions

Recurrence after a 5-year cancer-free period in patients with BC who underwent RC was infrequent. Severe sarcopenia was not associated with an increased recurrence risk but was associated with non-cancer-specific mortality. Thus, patients with severe sarcopenia may not need continuous surveillance after a 5-year cancer-free period.

## Figures and Tables

**Figure 1 cancers-15-01489-f001:**
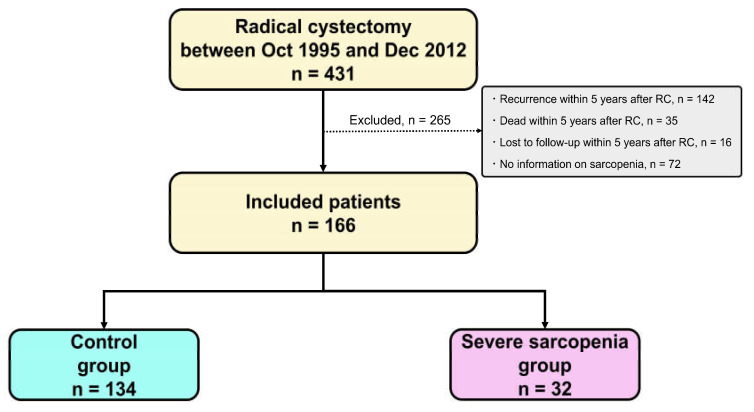
Patient selection. The numbers of patients included and excluded from the present study are shown. RC, radical cystectomy.

**Figure 2 cancers-15-01489-f002:**
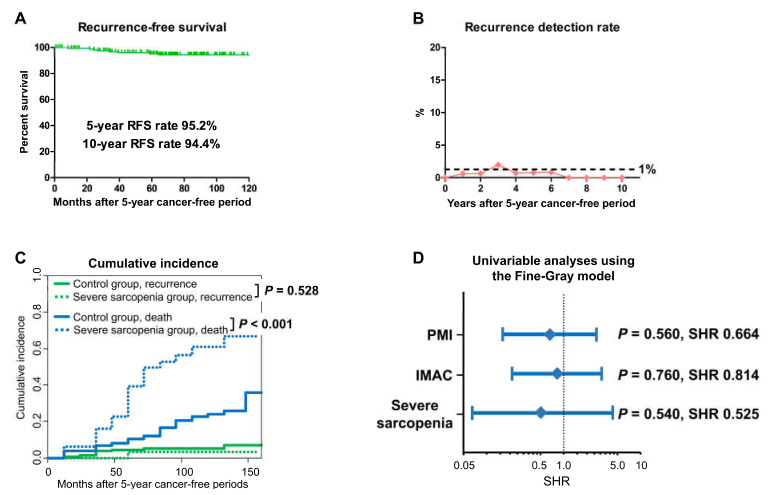
Recurrence-free survival (RFS) and recurrence detection rate. RFS in all included patients after a 5-year cancer-free period was evaluated using the Kaplan–Meier method (**A**). Recurrence detection rate ([number of patients with recurrence/number of patients with surveillance during a certain period] × 100) was evaluated (**B**). The Gray test was performed to compare cumulative incidences between the control and severe sarcopenia groups (**C**). Univariable analyses were performed to assess the impact of severe sarcopenia on recurrence, adjusting for the competing risk of death using the Fine-Gray subdistribution hazards model (**D**). PMI, psoas muscle index. IMAC, intramuscular adipose tissue content. SHR, subdistribution hazard ratio.

**Figure 3 cancers-15-01489-f003:**
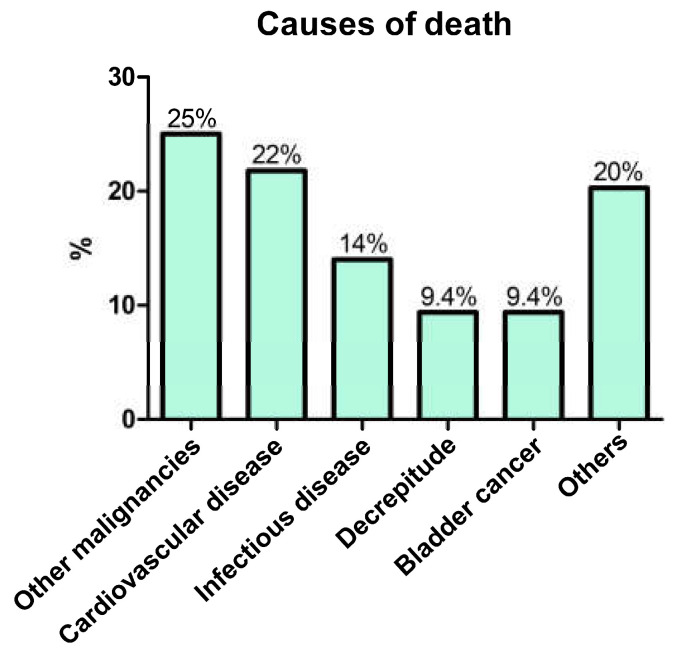
Causes of death after a 5-year cancer-free period.

**Figure 4 cancers-15-01489-f004:**
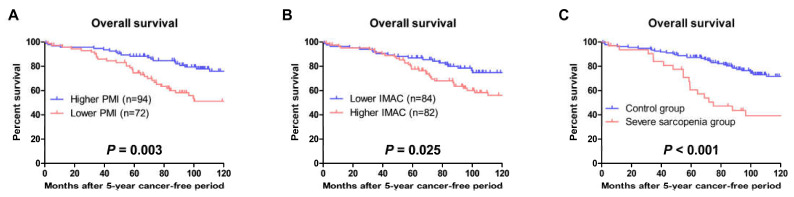
Overall survival. Overall survival after a 5-year cancer-free period was evaluated using the Kaplan–Meier method and compared using the log-rank test between patients with higher and lower psoas muscle index (PMI) values (**A**), lower and higher intramuscular adipose tissue content (IMAC) values (**B**), and patients in the control and severe sarcopenia groups (**C**).

**Figure 5 cancers-15-01489-f005:**
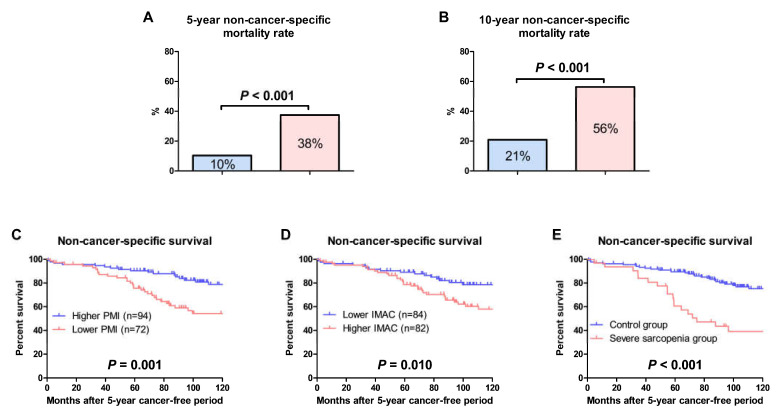
Non-cancer-specific survival. The 5-year (**A**) and 10-year (**B**) non-cancer-specific mortality rates were compared between the control and severe sarcopenia groups using the chi-squared test. Non-cancer-specific survival after a 5-year cancer-free period were evaluated using the Kaplan–Meier method and compared using the log-rank test between patients with higher and lower psoas muscle index (PMI) values (**C**), lower and higher intramuscular adipose tissue content (IMAC) values (**D**), and patients in the control and severe sarcopenia groups (**E**).

**Table 1 cancers-15-01489-t001:** Patients’ backgrounds.

	All, *n* = 166	Control Group, *n* = 134	Severe Sarcopenia Group, *n* = 32	*p* Value
Age, years	73 (64–77)	70 (63–77)	76 (73–81)	<0.001
Male	124 (75%)	99 (74%)	25 (78%)	0.620
ECOG PS ≥ 2	3 (1.8%)	0 (0.0%)	3 (9.4%)	0.007
Hypertension	71 (43%)	57 (43%)	14 (44%)	0.901
Diabetes mellitus	19 (11%)	15 (11%)	4 (13%)	0.765
Cardiovascular disease	40 (24%)	31 (23%)	9 (28%)	0.533
Chronic kidney disease	78 (47%)	61 (46%)	17 (53%)	0.439
Clinical stage				
cT3 or cT4	74 (45%)	58 (43%)	16 (50%)	0.492
cN1–3	16 (9.6%)	15 (11%)	1 (3.1%)	0.313
Neoadjuvant chemotherapy	85 (51%)	71 (53%)	14 (44%)	0.348
Ileal neobladder	109 (66%)	91 (68%)	18 (56%)	0.212
Pathological outcomes				
Pure urothelial carcinoma	133 (80%)	110 (82%)	23 (72%)	0.193
pT0	31 (19%)	25 (19%)	6 (19%)	0.990
pT3 or pT4	31 (19%)	24 (18%)	7 (22%)	0.605
pN1–3	11 (6.6%)	9 (6.7%)	2 (6.3%)	1.000
Grade 3	122 (74%)	101 (75%)	21 (66%)	0.262
Lymphovascular invasion	50 (30%)	42 (31%)	8 (24%)	0.482
Positive surgical margin	4 (2.4%)	4 (3.0%)	0 (0.0%)	1.000
Adjuvant chemotherapy	19 (11%)	14 (10%)	5 (16%)	0.372
Follow-up period, months	95 (66–135)	100 (71–137)	70 (49–125)	

All data are presented as *n* (%) or medians (interquartile ranges). ECOG PS, Eastern Cooperative Oncology Group performance status.

**Table 2 cancers-15-01489-t002:** Univariable analyses for non-cancer-specific survival.

	Factor	*p* Value	Hazard Ratio	95% CI
Age	Continuous	<0.001	1.110	1.069–1.154
Sex	Male	0.961	1.016	0.546–1.891
ECOG PS	≥2	0.001	8.127	2.490–26.52
Hypertension	Present	0.027	1.795	1.069–3.011
Diabetes mellitus	Present	0.891	0.942	0.404–2.198
History of cardiovascular disease	Positive	0.289	1.376	0.763–2.481
Chronic kidney disease	Present	<0.001	3.385	1.864–6.148
Neoadjuvant chemotherapy	Received	0.751	0.915	0.529–1.582
Histology	Pure UC	0.423	0.781	0.428–1.428
Pathological T stage	pT0	0.613	1.188	0.610–2.313
Pathological T stage	pT3 or pT4	0.619	0.840	0.424–1.666
Pathological N stage	pN1–3	0.952	0.969	0.348–2.696
Tumor grade	Grade 3	0.290	0.741	0.424–1.292
Lymphovascular invasion	Positive	0.574	0.847	0.475–1.511
Adjuvant chemotherapy	Received	0.113	0.470	0.185–1.195
Psoas muscle index	Lower than cut-off values	0.002	2.329	1.378–3.936
Intramuscular adipose tissue content	Higher than cut-off values	0.011	2.000	1.170–3.421
Severe sarcopenia	Present	<0.001	3.106	1.815–5.316

ECOG PS, Eastern Cooperative Oncology Group performance status; CI, confidence interval; UC, urothelial carcinoma.

**Table 3 cancers-15-01489-t003:** Multivariable analyses for non-cancer-specific survival.

**Model 1**	**Factor**	** *p* ** **Value**	**Hazard Ratio**	**95% CI**
Age	Continuous	<0.001	1.084	1.037–1.133
ECOG PS	≥2	0.066	3.266	0.923–11.56
Hypertension	Present	0.269	1.380	0.779–2.443
Chronic kidney disease	Present	0.017	2.156	1.148–4.049
Psoas muscle index	Lower than cut-off values	0.439	1.267	0.696–2.308
**Model 2**	**Factor**	** *p* ** **Value**	**Hazard Ratio**	**95% CI**
Age	Continuous	0.001	1.082	1.035–1.132
ECOG PS	≥2	0.055	3.415	0.973–11.99
Hypertension	Present	0.257	1.395	0.785–2.480
Chronic kidney disease	Present	0.016	2.178	1.156–4.103
Intramuscular adipose tissue content	Higher than cut-off values	0.367	1.377	0.688–2.757
**Model 3**	**Factor**	** *p* ** **Value**	**Hazard Ratio**	**95% CI**
Age	Continuous	<0.001	1.080	1.035–1.127
ECOG PS	≥2	0.193	2.365	0.647–8.643
Hypertension	Present	0.184	1.475	0.831–2.619
Chronic kidney disease	Present	0.014	2.195	1.175–4.100
Severe sarcopenia	Present	0.047	1.909	1.007–3.619

CI, confidence interval.

## Data Availability

The data can be shared up on request.

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
