# Peer review of "Combination of Muscle Quantity and Quality Is Useful to Assess the Necessity of Surveillance after a 5-Year Cancer-Free Period in Patients Who Undergo Radical Cystectomy: A Multi-Institutional Retrospective Study"

_cancers, 2023, doi:10.3390/cancers15051489_

Round 1
Reviewer 1 Report
Dear authors,
your study is really interesting as it investigates about a rarely studied but relevant clinical theme.
the paper is very well written, the results are sound and interesting.
these are my suggestion
line 70 remove regarding BC
in the discussion the authors should put more the focus on the fact that patient with severe sarcopenia were significatively older than control patients.
As you know, skeletal muscle loss is associated with ageing and therefore the fact that patients with higher sarcopenia are indeed the patients with older age may lead to an association bias.
thanks a lot
Author Response
First, we sincerely appreciate for the kind comments and suggestions of all reviewers and editors. We attached our revised manuscript here, as well as a point-by-point response to the comments.
Response to Reviewer
- line 70 remove regarding BC
Response: Thank you for the pointing comment. We agree with this comment. We revised the sentence as follows.
Page 2, line 66 (Introduction section)
Although sarcopenia has been reported to be associated with unfavorable prognosis in patients who underwent surgical treatment for various malignancies [6,7], regarding BC, the prognostic value of sarcopenia in patients who undergo RC remains controversial [8-11].
- In the discussion the authors should put more the focus on the fact that patient with severe sarcopenia were significatively older than control patients. As you know, skeletal muscle loss is associated with ageing and therefore the fact that patients with higher sarcopenia are indeed the patients with older age may lead to an association bias.
Response: Thank you for the pointing comment. We agree with this comment. As the reviewer pointed out, skeletal muscle loss is associated with ageing and patients in the severe sarcopenia group were significantly older than patients in the control group in the present study. Thus, it might have caused an association bias regardless of the adjustment for age in the multivariable analyses. We added the following sentences to the limitation section.
Page 11, line 338 (Limitation section)
In addition, skeletal muscle loss is associated with ageing and patients in the severe sarcopenia group were significantly older than patients in the control group in the present study (Table 1). Thus, it might have caused an association bias regardless of the adjustment for age in the multivariable analyses.
Reviewer 2 Report
The authors propose the use of the Combination of muscle quantity and quality to predict the need for follow-up in cystectomy patients with a negative follow-up of at least 5 years.
The authors conclude that severe sarcopenia does not correlate with the risk of recurrence in these patients but with the risk of non-cancer-related death at 10 years, and therefore suggest that patients with severe sarcopenia at 5 years after CR in the absence of recurrence can discontinue follow-up
I honestly do not fully understand the aim of this work. It is clear that sarcopenia status correlates with age and that age is a physiological predictor of overall mortality.
It is not easy to investigate what the ideal follow-up scheme is, especially when one wants to assess when the risk of death from other causes outweighs the risk of recurrence. To do this, some authors have used Weibull regression models that take into account the change of hazards over time.
I have a few considerations about this manuscript:
I suggest the authors, when investigating the risk of recurrence in patients with severe sarcopenia, perform a competing risk analysis considering the risk of non-cancer mortality as a competing risk event. As the authors report, and as can be imagined, the group of patients with severe sarcopenia is older and tends to die more this means that patients censored because they died are no longer at risk for the recurrence event. A competing risk analysis would therefore be more correct. Please see: Satagopan JM, Ben-Porat L, Berwick M, Robson M, Kutler D, Auerbach AD. A note on competing risks in survival data analysis. Br J Cancer. 2004;91(7):1229-1235. doi:10.1038/sj.bjc.6602102
It is unclear why in the case of recurrence risk, the authors use a cox regression analysis, whereas in the case of non-cancer survival, they use a logistic regression. Furthermore, the authors should explain why they give only results at 10 years. It would be better, in any case, to also perform a cox regression in the case of non-cancer-specific survival and then analyze the results at 10 years.
I also suggest reporting the concordance or C-index of the model instead of the ROC curve.
Other scores predict the risk of death from other causes, such as the Charlson comorbidity index; it would be interesting to compare the CCI with sarcopenia in this group of patients.
Has cancer-specific survival not been analyzed in view of the low number of events? if so, it should be added to the limitations of the work
Author Response
First, we sincerely appreciate for the kind comments and suggestions of all reviewers and editors. We attached our revised manuscript here, as well as a point-by-point response to the comments.
Response to Reviewer
- I suggest the authors, when investigating the risk of recurrence in patients with severe sarcopenia, perform a competing risk analysis considering the risk of non-cancer mortality as a competing risk event. As the authors report, and as can be imagined, the group of patients with severe sarcopenia is older and tends to die more this means that patients censored because they died are no longer at risk for the recurrence event. A competing risk analysis would therefore be more correct. Please see: Satagopan JM, Ben-Porat L, Berwick M, Robson M, Kutler D, Auerbach AD. A note on competing risks in survival data analysis. Br J Cancer. 2004;91(7):1229-1235. doi:10.1038/sj.bjc.6602102
Response: Thank you for the pointing comment. We agree with this comment. The cumulative incidences of recurrence was estimated and death before recurrence was defined as a competing risk. The Gray test was performed to compare cumulative incidences between the control and severe sarcopenia groups. Subsequent univariable analyses were performed to assess the impact of severe sarcopenia on recurrence, adjusting for the competing risk of death using the Fine-Gray subdistribution hazards model. Results showed that the cumulative incidences rate of recurrence was not significantly different between the control and severe sarcopenia groups (revised Fig. 2C) and none of lower PMI, higher IMAC, and severe sarcopenia showed significantly higher probabilities of recurrence (revised Fig. 2D). We revised the manuscript as follows.
Page 1, line 42 (Abstract section)
Univariable analyses were performed to assess the impact of severe sarcopenia on recurrence, adjusting for the competing risk of death using the Fine-Gray competing risk regression model.
Page 2, line 47 (Abstract section)
In the Fine-Gray competing risk regression model, severe sarcopenia did not show a significant higher probability of recurrence, with an adjusted subdistribution hazard ratio of 0.525 (P = 0.540), whereas severe sarcopenia was significantly associated with non-cancer-specific survival (HR 1.909, P = 0.047).
Page 4, line 160 (Methods section)
Moreover, the cumulative incidences of recurrence was estimated and death before recurrence was defined as a competing risk. The Gray test was performed to compare cumulative incidences between the control and severe sarcopenia groups. Subsequent univariable analyses were performed to assess the impact of severe sarcopenia on recurrence, adjusting for the competing risk of death using the Fine-Gray subdistribution hazards model.
Page 6, line 206 (Results section)
In the Gray test, the cumulative incidence rate of recurrence was not significantly different between the control and severe sarcopenia groups (Fig. 2C, P = 0.529).
Page 6, line 210 (Results section)
Similarly, none of lower PMI, higher IMAC, and severe sarcopenia showed significant higher probabilities of recurrence (Fig. 2D; subdistribution hazard ratio [SHR] 0.664, 95% confidence interval [CI] 0.168–2.630, P = 0.560; SHR 0.814, 95% CI 0.220–3.010, P = 0.760; SHR 0.525, 95% CI 0.067–4.140, P = 0.540; respectively).
Page 6, line 218 (Figure legends section)
The Gray test was performed to compare cumulative incidences between the control and severe sarcopenia groups (C). Univariable analyses were performed to assess the impact of severe sarcopenia on recurrence, adjusting for the competing risk of death using the Fine-Gray subdistribution hazards model (D). PMI, psoas muscle index. IMAC, intramuscular adipose tissue content. SHR, subdistribution hazard ratio.
- It is unclear why in the case of recurrence risk, the authors use a cox regression analysis, whereas in the case of non-cancer survival, they use a logistic regression. Furthermore, the authors should explain why they give only results at 10 years. It would be better, in any case, to also perform a cox regression in the case of non-cancer-specific survival and then analyze the results at 10 years. I also suggest reporting the concordance or C-index of the model instead of the ROC curve.
Response: Thank you for the pointing comment. We agree with this comment. The editor has also pointed out at the preliminary review. According to the editor’s and reviewer’s suggestions, we used Cox proportional hazards regression models instead of logistic regression models to identify the significant factors associated with non-cancer-specific survival. Accordingly, we reevaluated optimal cut-off values of the psoas muscle index (PMI) and intramuscular adipose tissue content (IMAC) for non-cancer-specific survival using receiver operating characteristic curves. Of 166 patients, 32 were diagnosed with severe sarcopenia. Regarding non-cancer-specific survival and overall survival, no significant change in the results was observed after reanalysis. We extensively revised the manuscript, figures, and tables. Moreover, we deleted the predictive analysis from the manuscript, considering manuscript length, small number of patients with severe sarcopenia, and its less impact in the present study.
Page 1, line 44 (abstract section)
The impact of severe sarcopenia on recurrence-free survival (RFS) and non-cancer-specific survival was evaluated using univariable and multivariable analyses.
Page 1, line 46 (abstract section)
Of 166 patients, 32 were diagnosed with severe sarcopenia. The 10-year RFS rate was 94.4%. In the Fine-Gray competing risk regression model, severe sarcopenia did not show a significant higher probability of recurrence, with an adjusted subdistribution hazard ratio of 0.525 (P = 0.540), whereas severe sarcopenia was significantly associated with non-cancer-specific survival (HR 1.909, P = 0.047).
Page 2, line 76 (Introduction section)
The aim of the present study was to evaluate the impact of low muscle quantity and quality on recurrence-free survival (RFS) and non-cancer-specific survival after a 5-year cancer-free period in patients with BC who underwent RC.
Page 4, line 166 (Methods section)
Univariable and multivariable Cox proportional hazards regression analyses were performed to evaluate the impact of severe sarcopenia on non-cancer-specific survival.
Page 5, line 184 (Results section)
The optimal cut-off values of PMI for non-cancer-specific mortality in men and women was 5.28 cm2/m2 and 6.35 cm2/m2, respectively. Of the 166 patients, 94 (57%) and 72 (43%) had PMI values higher and lower than the cut-off values, respectively.
Page 5, line 187 (results section)
The optimal cut-off values of IMAC for non-cancer-specific mortality in men and women was -0.49 and -0.04, respectively. Of the 166 patients, 84 (51%) and 82 (49%) had IMAC values lower and higher than the cut-off values, respectively.
Page 5, line 191 (results section)
Patients were divided into two groups: those with lower PMI and higher IMAC values (severe sarcopenia group, n = 32) and those with higher PMI and/or lower IMAC values (control group, n = 134) (Fig. 1).
Page 6, line 209 (results section)
Similarly, lower PMI, higher IMAC, and severe sarcopenia were not significantly associated with shorter RFS (Fig. 2D; hazard ratio [HR] 0.818, 95% confidence interval [CI] 0.202–3.320, P = 0.779; HR 0.872, 95% CI 0.233–3.261, P = 0.839; HR 0.680, 95% CI 0.084–5.518, P = 0.718; respectively).
Page 7, line 223 (Results section)
The main causes of death during the entire follow-up period were other malignancies (25%) and CVD (22%), followed by infectious diseases (14%) (Fig. 3). Of the 64 patients who died from any cause, six (9.4%) died of BC (Fig. 3).
Page 7, line 229 (Results section)
The OS in patients with lower PMI values was significantly shorter than that in patients with higher PMI values (Fig. 4A, P = 0.003). The OS in patients with higher IMAC values was significantly shorter than that in patients with lower IMAC values (Fig. 4B, P = 0.025).
Page 7, line 241 (Results section)
The 5-year and 10-year non-cancer-specific mortality rates in the severe sarcopenia group were significantly higher than those in the control group (Fig. 5A and 5B; 38% vs. 10%, P < 0.001; 56% vs. 21%, P < 0.001; respectively). The non-cancer-specific survival in patients with lower PMI values was significantly shorter than that in patients with higher PMI values (Fig. 5C, P = 0.001). The non-cancer-specific survival in patients with higher IMAC values was significantly shorter than that in patients with lower IMAC values (Fig. 5D, P = 0.010). The non-cancer-specific survival in the severe sarcopenia group was significantly shorter than that in the control group (Fig. 5E, P < 0.001). In univariable analyses, age, EGOS PS, HTN, CKD, lower PMI, higher IMAC, and severe sarcopenia were significantly associated with shorter non-cancer-specific survival (Table 2). In multivariable analyses, lower PMI and higher IMAC were not significantly associated with shorter non-cancer-specific survival (HR 1.267, 95% CI 0.696–2.308, P = 0.439; HR 1.377, 95% CI 0.688–2.757, P = 0.367; respectively), whereas severe sarcopenia was significantly associated with shorter non-cancer-specific survival (HR 1.909, 95% CI 1.007–3.619, P = 0.047) (Table 3).
- Other scores predict the risk of death from other causes, such as the Charlson comorbidity index; it would be interesting to compare the CCI with sarcopenia in this group of patients.
Response: Thank you for the pointing comment. We agree with this comment. However, unfortunately, our data did not include the information on the CCI. Our future study will address this issue.
- Has cancer-specific survival not been analyzed in view of the low number of events? if so, it should be added to the limitations of the work.
Response: Thank you for the pointing comment. We agree with this comment. As the reviewer mentioned, cancer-specific survival was not evaluated because of small number of event (6 events). We added the following sentence to the limitation section.
Page 11, line 343 (Limitation section)
Moreover, the small number of cancer-specific death prevented us from evaluating cancer-specific survival.